# Nonasymptotic Laplace approximation under model misspecification

## Abstract

In this note, we present non-asymptotic two-sided bounds to the log-marginal likelihood in Bayesian inference. The classical Laplace approximation is recovered as the leading term. Our derivation permits model misspecification and allows the parameter dimension to grow with the sample size. We do not make any assumptions about the asymptotic shape of the posterior, and instead require certain regularity conditions on the likelihood ratio and that the posterior is sufficiently concentrated. We envision the derived bounds to be widely applicable in establishing model selection consistency of Bayesian procedures in non-conjugate settings, especially when the true model potentially lies outside the class of candidate models considered.

## 1 Introduction

Suppose data $Y$ is modeled according to a probability distribution $\mathbb{P}_\theta$, with the parameter space $\Theta \subseteq \Re^d$ a closed convex set. For each $\theta$, suppose $\mathbb{P}_\theta$ admits a density $p_\theta = (d\mathbb{P}_\theta/d\mu)$ with respect to a common $\sigma$-finite measure $\mu$ on the sample space $\mathcal{Y}$. Assume the map $(y, \theta) \mapsto p_\theta(y)$ is jointly measurable, and let $\ell(\theta) = \log p_\theta(Y)$ be the log-likelihood function. Let $\pi(\cdot)$ be a continuous proper prior on $\Theta$ and let $\gamma(\cdot)$ denote the corresponding posterior distribution so that for any measurable set $B$,

$$\gamma(B) = \frac{\int_B e^{\ell(\theta)} \, \pi(\theta)d\theta}{\mathcal{Z}_\gamma}, \quad \mathcal{Z}_\gamma = \int_\Theta e^{\ell(\theta)} \, \pi(\theta)d\theta. \tag{1}$$

The posterior normalizing constant $\mathcal{Z}_\gamma$ in (1) is commonly referred to as the marginal likelihood or evidence (Robert, 2007). The marginal likelihood is an ubiquitous tool for model comparison and selection in Bayesian statistics as it encapsulates an automatic penalty for model complexity.

Barring conjugate settings, the multivariate integral in (1) is rarely available in closed form, necessitating approximations to the marginal likelihood for computation as well as theoretical analysis. Laplace's integral approximation method, commonly referred to as the Laplace approximation (Tierney & Kadane, 1986), is arguably the most well-known and widely used approximation; see Robert (2007); Ghosh et al. (2007) for book level treatments. In regular parametric models with $n$ independent and identically distributed samples and $\widehat{\theta}$ the maximum likelihood estimator, the Laplace approximation takes the form $\log \mathcal{Z}_\gamma \approx \ell(\widehat{\theta}) - d \log n/2$. The quantity on the right hand side is, up to a scale factor, the celebrated Bayesian information criterion (Schwarz, 1978), which is thus realized as an asymptotic approximation to the log-marginal likelihood. Throughout the article, we reserve the phrase Laplace approximation to exclusively refer to the above and not the closely related problem of approximating posterior expectations of functionals (Tierney & Kadane, 1986; Tierney et al., 1989; Miyata, 2004; Ruli et al., 2016).

The usual heuristic derivation of the Laplace approximation proceeds by performing a Taylor series expansion of the log-likelihood function on a neighborhood of the maximum likelihood estimator or the posterior mode to reduce the integral to a Gaussian integral. This argument can be made rigorous (Chen, 1985; Kass et al., 1990) under the assumptions of a Bernstein–von Mises theorem guaranteeing the posterior assuming a Gaussian shape asymptotically; see also Remark 1.4.5. of Ghosh & Ramamoorthi (2003) for an exposition

along these lines. Shun & McCullagh (1995) showed that if the dimension is comparable with the sample size, then the usual Laplace approximation is not valid.

In this article, we present a derivation of the Laplace approximation without assuming an asymptotic Gaussian shape of the posterior. Specifically, we obtain non-asymptotic two-sided bounds on $\log \mathcal{Z}_\gamma$ with the same leading term, valid with high probability under the true data distribution. While the assumption of a true data generating distribution is standard in existing derivations (Kass et al., 1990; Cavanaugh & Neath, 1999), we refrain from assuming the model to be correctly specified. In such misspecified settings, the parameter value in the model class closest to the true distribution in Kullback–Leibler divergence assumes the role of the true parameter in well-specified settings.

Our derivation crucially exploits the concentration of the posterior distribution (Kleijn & van der Vaart, 2006) around this pseudo-true parameter, which typically requires milder assumptions compared to asymptotic normality. For example, even in the linear regression setup, one needs strong prior flatness conditions for asymptotic normality when the parameter dimension grows with the sample size (Bontemps, 2011). We show that the concentration phenomenon is sufficient to localize the assumptions on the likelihood surface on a neighborhood around the pseudo-true parameter, unlike the global assumptions in Cavanaugh & Neath (1999). We verify our conditions in the setting of a generalized linear model with growing parameter dimension, and the same template can be used in other settings such as quantile regression and more generally, for model selection beyond the Gaussian linear model (Rossell & Rubio, 2018).

## 2 Main result

As noted in the introduction, we operate in misspecified framework allowing the true data distribution $\mathbb{P}$ to lie outside the model class $\{\mathbb{P}_\theta : \theta \in \Theta\}$. Without loss of generality, assume $\mathbb{P} \ll \mu$ and let $p(\cdot) = d\mathbb{P}/d\mu(\cdot)$. We shall reserve the symbol $\mathbb{E}$ to denote an expectation with respect to $\mathbb{P}$. Let

$$\theta^* = \arg\min_{\theta \in \Theta} D(p \,||\, p_\theta) = \arg\max_{\theta \in \Theta} \mathbb{E}\ell(\theta) \tag{2}$$

be the closest Kullback–Leibler point to the truth inside the parameter space, with $D(p \,||\, q) = E_p(\log p/q)$ the Kullback–Leibler divergence between densities $p$ and $q$. In a misspecified setting, the pseudo-true parameter $\theta^*$ plays the role of the true parameter in well-specified models.

We now lay down the assumptions underlying our main result. For any $\theta, \theta^\dagger \in \Theta$, we let $\ell(\theta, \theta^\dagger) = \ell(\theta) - \ell(\theta^\dagger)$ denote the log-likelihood ratio. Throughout $C, C_1, C_2, \ldots$ denote global positive constants. Let $\ell_r(\theta) = \ell(\theta) - \mathbb{E}\ell(\theta)$ and $B^* \equiv B^*_{W,R} = \{\theta \in \Theta : (\theta - \theta^*)^{\mathrm{T}} W (\theta - \theta^*) \leq R\,d\}$ for a fixed positive definite matrix $W$ and a constant $R > 0$.

**Assumption 1** (Likelihood ratio: deterministic part)**.** *There exists a fixed $d \times d$ positive definite matrix $H$ and a constant $c \in (1/2, 1)$ such that for all $\theta \in B^*$,*

$$(\theta - \theta^*)^{\mathrm{T}} H (\theta - \theta^*)/(2c) \geq -\mathbb{E}\,\ell(\theta, \theta^*) \geq (\theta - \theta^*)^{\mathrm{T}} H (\theta - \theta^*)/2. \tag{3}$$

**Assumption 2** (Likelihood ratio: stochastic part)**.** *There exists a positive constant $C$ and $\tilde{\delta} \in (0, 1/4)$ such that $\mathbb{P}\big\{ \sup_{\theta \in B^*} |\ell_r(\theta) - \ell_r(\theta^*)| \leq C\,d \big\} \geq 1 - \tilde{\delta}$.*

**Assumption 3** (Prior)**.** *The prior distribution $\pi$ is continuous and nowhere zero on $\Theta$.*

**Assumption 4** (Posterior concentration)**.** *There exists constants $\eta, \delta \in (0, 1/4)$ such that $\mathbb{P}\big\{\gamma(B^*) \geq 1 - \eta\big\} \geq 1 - \delta$.*

Assumptions 1 and 2 together posit conditions on the log-likelihood ratio $\ell(\theta, \theta^*)$ on a neighborhood $B^*$ of $\theta^*$. We separate the conditions into stochastic and deterministic components by writing $\ell(\theta, \theta^*) = \mathbb{E}\ell(\theta, \theta^*) + \ell_r(\theta) - \ell_r(\theta^*)$.

Assumption 1 posits that $-\mathbb{E}\,\ell(\theta, \theta^*)$ can be approximated by a quadratic form in $(\theta - \theta^*)$ in a local neighborhood of $\theta^*$. This is a standard assumption in parametric models; see, e.g. Spokoiny (2012a). If $\theta \mapsto \mathbb{E}\ell(\theta)$ is twice differentiable, a natural choice to find $H$ is to perform a Taylor expansion. Since $\nabla\mathbb{E}\ell(\theta^*) = 0$,

the condition (3) is satisfied if $c^{-1}H \succsim -\nabla^2 \mathbb{E}\ell(\theta) \succsim H$ for all $\theta \in B^*$, where $A_1 \succsim A_2$ denotes $A_1 - A_2$ is nonnegative definite. Thus in well-specified regular models, the matrix $H$ plays the role of the Fisher Information matrix. Another particular simplification arises for well-specified models where $-\mathbb{E}\ell(\theta, \theta^*)$ is the Kullback–Leibler divergence $D(p_{\theta^*} \,\|\, p_\theta)$, which is known to be locally equivalent to a weighted Euclidean metric in many parametric models.

Assumption 2 requires control over the supremum of the centered empirical process $\ell_r(\theta)$ as $\theta$ varies over the set $B^*$. In specific examples, this can be achieved by first bounding the expected supremum $\mathbb{E}\sup_{\theta \in B^*} \ell_r(\theta) - \ell_r(\theta^*)$ using a standard chaining argument and then use a concentration inequality for the supremum around its expectation. Refer to Talagrand (2006); Boucheron et al. (2013); Vershynin (2018) for such arguments for general empirical processes and van de Geer (2006); Spokoiny (2012b) for a more specialized statistical context. We also mention the more recent work (Dirksen, 2015) which directly obtains a high-probability bound for the supremum of an empirical process using generic chaining. Some smoothness assumption on the likelihood surface is necessary to apply these results, which may be posed on the increments or alternatively, on the gradient, of the likelihood process. We provide some specific examples in the next section.

Assumption 3 is broadly satisfied and Assumption 4 requires the posterior distribution $\gamma(\cdot)$ to place sufficient mass around the pseudo-true parameter $\theta^*$. A set of general conditions for posterior concentration in misspecified models can be found in Kleijn & van der Vaart (2006); see also De Blasi & Walker (2013); Sriram et al. (2013); Ramamoorthi et al. (2015); Atchadé (2017); Bhattacharya et al. (2019). We prove a general theorem for misspecified high-dimensional generalized linear models in the Appendix. With these ingredients in place, we state a two-sided bound on the log-marginal likelihood in Theorem 1 below.

**Theorem 1.** *Recall $\mathcal{Z}_\gamma$ from (1), and assume Assumptions 1–4 are satisfied. Then, with $\mathbb{P}$-probability at least $(1 - \delta - \tilde{\delta})$, the following bounds in (4) and (5) hold:*

$$\log \mathcal{Z}_\gamma \leq \ell(\theta^*) - \frac{\log|H|}{2} + \left[ C_1 d + \log\left\{ \frac{\sup_{\theta \in B^*} \pi(\theta)}{1 - \eta} \right\} + \log P(\|\xi\|^2 \leq Rd) \right], \tag{4}$$

*where $C_1 = C + \log(2\pi)/2$ and $\xi \sim \mathcal{N}_d(0, W^{1/2}H^{-1}W^{1/2})$. Also,*

$$\log \mathcal{Z}_\gamma \geq \ell(\theta^*) - \frac{\log|H|}{2} + \left\{ C_2 d + \log \inf_{\theta \in B^*} \pi(\theta) + \log P(\|\xi\|^2 \leq Rc^{-1}d) \right\}, \tag{5}$$

*where $C_2 = -C + \log(2\pi)/2 + c/2$ and $\xi$ is the same as before.*

A proof of Theorem 1 is provided in the Appendix. An inspection of the proof will reveal that the concentration of the posterior in Assumption 4 is only utilized for the upper bound. Some additional remarks regarding the result are in order. We state the bounds in terms of $\ell(\theta^*)$, and not $\ell(\widehat{\theta})$, for convenience of theoretical analysis. When comparing models, this helps to get rid of one layer on randomness stemming from the respective $\widehat{\theta}$ for each model. It is straightforward to modify the argument and state the bounds in terms of $\ell(\widehat{\theta})$ as detailed in the proof. In regular parametric models with $n$ independent and identically distributed samples, $|H| \asymp n^{-d/2}$, leading to the recognizable $-d \log n/2$ penalty in the Bayesian information criterion. Lv & Liu (2014) defined a generalized Bayesian information criterion for misspecified models with an additional term containing the sandwich covariance appearing in the asymptotic distribution of the maximum likelihood estimator under misspecification. However, the sandwich covariance term does not appear in the asymptotic limit of the posterior under misspecification (Kleijn & Van der Vaart, 2012), and also does not show up in our calculations.

## 3 Verification of assumptions

In this section, we verify the Assumptions in §2 for a generalized linear model, which subsumes a wide array of examples encountered in practice. For a more direct approach for the special case of i.i.d. exponential family models, refer to Schwarz (1978); Haughton (1988). Consider covariate-response pairs $\{(y_i, x_i)\}_{i=1}^n$ with $y_i \in \Re$ and $x_i \in \Re^d$. We consider the moderately high-dimensional regime where $d$ is less than $n$, but allowed to grow with $n$. Let $y = (y_1, \ldots, y_n)^{\mathrm{T}}$ and let $X$ denote the $n \times d$ matrix of covariates. We assume a model on $y_i$ conditional on the covariates $x_i$ independently according to a generalized linear model

$P_{x_i^\mathrm{T}\beta}$ in canonical form, with the log-likelihood $\ell(\beta) = \log p_\beta(y) = \sum_{i=1}^n \left\{y_i x_i^\mathrm{T}\beta - a(x_i^\mathrm{T}\beta)\right\}$, where $\beta \in \Re^d$ is the unknown vector of regression parameters. The function $a : \Re \to \Re$ is convex; we shall denote its first and second derivatives by $a^{(1)}$ and $a^{(2)}$ respectively. We operate in a misspecified framework and do not assume the existence of a true regression parameter, and instead only make tail assumptions on the true data distribution. The pseudo-true parameter $\beta^*$ satisfies

$$\nabla\mathbb{E}\ell(\beta^*) = \sum_{i=1}^n \{\mathbb{E}y_i - a^{(1)}(x_i^\mathrm{T}\beta^*)\}x_i = 0_d. \tag{6}$$

### 3.1 Verification of Assumptions 1 and 2

Let us consider Assumption 1 first. We have,

$$
\begin{aligned}
-\mathbb{E}\ell(\beta, \beta^*) &= \sum_{i=1}^n \left\{ a(x_i^\mathrm{T}\beta) - a(x_i\beta^*) - x_i^\mathrm{T}(\beta - \beta^*)\, a^{(1)}(x_i^\mathrm{T}\beta^*) \right\} \\
&= \frac{1}{2}(\beta - \beta^*)^\mathrm{T}\left\{ \sum_{i=1}^n a^{(2)}(x_i^\mathrm{T}\tilde\beta)\, x_i x_i^\mathrm{T} \right\}(\beta - \beta^*),
\end{aligned}
$$

for some $\tilde\beta$ in the line segment joining $\beta$ and $\beta^*$. The first equality in the above display utilizes the identity (6). Letting $u_i^2 = \inf_{\beta \in B^*} a^{(2)}(x_i^\mathrm{T}\beta)$ and $v_i^2 = \sup_{\beta \in B^*} a^{(2)}(x_i^\mathrm{T}\beta)$ for $i = 1, \ldots, n$, we have $(\beta - \beta^*)^\mathrm{T}\left\{\sum_{i=1}^n u_i^2 x_i x_i^\mathrm{T}\right\}(\beta - \beta^*) \le -\mathbb{E}\ell(\beta, \beta^*) \le (\beta - \beta^*)^\mathrm{T}\left\{\sum_{i=1}^n v_i^2 x_i x_i^\mathrm{T}\right\}(\beta - \beta^*)^\mathrm{T}$ for all $\beta \in B^*$. Thus, we can set $H = \sum_{i=1}^n u_i^2 x_i x_i^\mathrm{T}$ and $c = \min_i u_i^2/v_i^2$ to satisfy Assumption 1.

The quantity $\ell_r(\beta) - \ell_r(\beta^*)$ appearing in Assumption 2 equals $\langle y - \mathbb{E}y, X(\beta - \beta^*)\rangle$ in the present context. Define an index set $\mathcal{T} = \{x \in \Re^d : \|x\| \le 1\}$, and a stochastic process $Z_\alpha = \langle y - \mathbb{E}y, X\alpha\rangle$ for $\alpha \in \mathcal{T}$. Observe that for any $\beta \ne \beta^* \in B^*$,

$$
\begin{aligned}
\left|\langle y - \mathbb{E}y, X(\beta - \beta^*)\rangle\right| &= \left|\langle y - \mathbb{E}y, \frac{X(\beta - \beta^*)}{\|\beta - \beta^*\|}\rangle\right| \|\beta - \beta^*\| \\
&\le \left(\sup_{\alpha \in \mathcal{S}^{d-1}} |\langle y - \mathbb{E}y, X\alpha\rangle|\right) R\left(\frac{d}{n}\right)^{1/2},
\end{aligned}
$$

where $\mathcal{S}^{d-1} = \{x \in \Re^d : \|x\| = 1\}$. Letting $\alpha_0 = 0_d$, we can thus bound $\sup_{\beta \in B^*} |\ell_r(\beta) - \ell_r(\beta^*)| \le R(d/n)^{1/2}\left(\sup_{\alpha \in \mathcal{T}} |Z_\alpha - Z_{\alpha_0}|\right)$. The verification of Assumption 2 thus requires control over the supremum of the stochastic process $(Z_\alpha)$, which in turn depends on the moment assumptions on the true data distribution. We illustrate this through two different examples below.

As a first example, assume that $(y - \mathbb{E}y)$ is a centered sub-Gaussian random variable (Vershynin, 2018), that is, there exists a constant $\tau > 0$ such that for any $v \in \Re^n$, $\mathbb{E}\exp\langle y - \mathbb{E}y, v\rangle \le \exp(\tau^2\|v\|^2/2)$. If the coordinates $y_i$ are independent, one may take $\tau = \max_i \|y_i - \mathbb{E}y_i\|_{\psi_2}$ to be the maximum of the sub-Gaussian norms of $(y_i - \mathbb{E}y_i)$; see Vershynin (2018) for definition of the sub-Gaussian norm $\|\cdot\|_{\psi_2}$. However, independence is not necessary for the above condition to hold and it can be verified for various dependence structures. In particular, if $y$ has a joint Gaussian distribution, then $\tau$ equals the largest eigenvalue of $\mathrm{cov}(y)$. Under the above sub-Gaussian assumption, the process $(Z_\alpha)$ has sub-Gaussian increments, since for any $\lambda \in \Re$,

$$\mathbb{E}e^{\lambda(Z_\alpha - Z_{\tilde\alpha})} \le e^{\lambda^2\tau^2\|X\alpha - X\tilde\alpha\|^2/2} \le e^{\lambda^2\tau^2\|X\|_2^2\|\alpha - \tilde\alpha\|^2},$$

where $\|X\|_2$ is the operator norm of $X$. For processes with sub-Gaussian increments, a convenient high-probability bound for the supremum was developed in Liaw et al. (2017, Theorem 4.1) as a corollary to the more general tail bound of Dirksen (2015). In preparation for applying their bound, we have $\|Z_\alpha - Z_{\tilde\alpha}\|_{\psi_2} \le \tau\|X\|_2\|\alpha - \tilde\alpha\|$ for any $\alpha, \tilde\alpha \in \mathcal{T}$. Also, $\mathrm{diam}(\mathcal{T}) = \sup_{\alpha, \tilde\alpha \in \mathcal{T}} \|\alpha - \tilde\alpha\| \le 2$ and the Gaussian width of $\mathcal{T}$, $\mathbb{E}\sup_{\alpha \in \mathcal{T}}\langle g, \alpha\rangle$ for $g \sim \mathcal{N}_d(0, I_d)$, is in the order of $d^{1/2}$. Thus, with probability at least $1 - e^{-d}$, $\sup_{\alpha \in \mathcal{T}} |Z_\alpha - Z_{\alpha_0}| \le C\tau\|X\|_2 d^{1/2}$. It then follows that with probability at least $1 - e^{-d}$, $\sup_{\beta \in B^*} |\ell_r(\beta) - \ell_r(\beta^*)| \le Cd$.

Alternatively, suppose $(y_i - \mathbb{E}y_i)$ are independent sub-exponential (Vershynin, 2018) random variables, so that there exist $g_i > 0$ and $\nu_i$ such that

$$\mathbb{E}e^{\lambda(y_i - \mathbb{E}y_i)} \le e^{\lambda^2 \nu_i^2/2}, \quad |\lambda| < g_i, \quad i = 1, \ldots, n.$$

Fix $\lambda$ such that $|\lambda| \le \min_i g_i := \bar{g}^{-1}$. Under the above assumption, we have, for any $\alpha, \tilde{\alpha} \in \mathcal{T}$ that

$$\mathbb{E}e^{\lambda \frac{Z_\alpha - Z_{\tilde{\alpha}}}{\|X\alpha - X\tilde{\alpha}\|}} = \prod_{i=1}^{n} \mathbb{E}e^{\lambda \frac{x_i^{\mathrm{T}}(\alpha - \tilde{\alpha})}{\|X\alpha - X\tilde{\alpha}\|}(y_i - \mathbb{E}y_i)} \le e^{\lambda^2 \sum_{i=1}^{n} \frac{\nu_i^2 \{x_i^{\mathrm{T}}(\alpha - \tilde{\alpha})\}^2}{\|X\alpha - X\tilde{\alpha}\|^2}}$$

$$\le e^{\lambda^2 \nu^2/2} \le e^{\lambda^2 \nu^2/\{2(1 - |\lambda|\bar{g})\}},$$

where $\nu = \max_i \nu_i$. From the second to the third step, we used that $|x_i^{\mathrm{T}}(\alpha - \tilde{\alpha})|/\|X\alpha - X\tilde{\alpha}\| \le 1$. Hence $Z_\alpha$ is a centered process on $\mathcal{T}$ with sub-exponential increments. Define a norm $d(\alpha_1, \alpha_2) = \|X\alpha_1 - X\alpha_2\|$. Clearly, $d(\alpha_1, \alpha_2) \le \|X\|_2$ for $\alpha_1, \alpha_2 \in \mathcal{T}$. From Theorem 2.1 of Baraud (2010),

$$\mathbb{P}\left[\sup_{\alpha \in \mathcal{T}} |Z_\alpha - Z_{\alpha_0}| > \|X\|_2\sqrt{1 + x} + \bar{g}x\right] \le 2e^{-x}, x > 0,$$

thereby verifying Assumption 2 by setting $x = d$.

## 3.2  Verification of Assumptions 3 and 4

Although literature on posterior contraction of regression parameters in linear models is abundant, both in moderately high-dimensional and ultra-high dimensional settings; see the introduction of Gao et al. (2015) for a general list of references; analogous results for generalized linear models are comparatively sparse, with the exception of Jiang et al. (2007). However, special cases including high dimensional logistic regression using a pseudo likelihood (Atchadé, 2017) and high-dimensional logistic regression using shrinkage priors (Wei & Ghosal, 2020) are available. Although it is possible to use such results directly to verify Assumption 4, this would typically come with additional restriction necessitated by the specific goals targeted in these papers. Jiang et al. (2007) operated in a well-specified setting where the use of a Gaussian prior leads to a restrictive assumption on the growth of the true coefficients; refer to the assumptions of Theorem 1 in pg. 1493. Atchadé (2017) considered a Laplace-type prior for the coefficients which obviated the need for such a restriction, but their results are specific to logistic regression. We focus on extending the result of Atchadé (2017) to accommodate other families and allow for model misspecification in the moderately high-dimensional regime with no sparsity assumption on the coefficients. We prove this result (Theorem 2) in the C; a sketch of the main ingredients is given below.

A posterior contraction result requires *non-local* versions of Assumptions 1 and 2 in the complement of the neighborhood under consideration. A fundamental technique (Ghosal et al., 2000) to prove such a result is to enforce that the likelihood ratio is appropriately small in $(B^*)^c$ and that the prior assigns sufficient probability around the true parameter in $B^*$. The first condition ensures that the numerator of the posterior probability of $(B^*)^c$ is small and the second condition prevents the denominator from becoming too small.

The separation of the likelihood in $(B^*)^c$ relies on the decomposition $\ell(\beta, \beta^*) = \ell_r(\beta) - \ell_r(\beta^*) + \mathbb{E}\ell(\beta, \beta^*)$ and then ensuring that $\mathbb{E}\ell(\beta, \beta^*)$ is sufficiently negative to offset the stochastic variation in $\ell_r(\beta) - \ell_r(\beta^*)$. Although $\mathbb{E}\ell(\beta, \beta^*)$ has a local quadratic shape for any member of the generalized linear model in $B^*$, $\mathbb{E}\ell(\beta, \beta^*)$ fails to be so outside $B^*$ for certain members in the family. For instance, $\mathbb{E}\ell(\beta, \beta^*)$ is approximately linear outside $B^*$ for logistic regression. Hence a suitable modification to the lower bound in Assumption 1 in required. This can be encapsulated through an assumption on $a$ as $a(t + h) \ge a(t) + h\,a^{(1)}(t) + \mathrm{r}(|h|)\,a^{(2)}(t)/2$ for all $t, h$, where $\mathrm{r}(\cdot)$ is a *rate function* (Atchadé, 2017) from $\mathbb{R}^+$ to $\mathbb{R}^+$ satisfying i) $\mathrm{r}(0) = 0$, ii) $\lim_{h \to 0} \mathrm{r}(h) = 0$ and iii) $r(h) \ge h^2/(\mathrm{r}_1 + \mathrm{r}_2 h)$ for $\mathrm{r}_1, \mathrm{r}_2 \ge 0$ not simultaneously 0. This class of $a$ functions includes the Gaussian $a(t) = t^2, \mathrm{r}(h) = h^2$; logistic $a(t) = -\log(1 + e^{-t}), \mathrm{r}(h) = h^2/(h + 2)$; and Poisson $a(t) = e^t, \mathrm{r}(h) = h^2$, among others. Using such a lower bound on $a$ it is possible to develop sharp lower bounds for $-\mathbb{E}\ell(\beta, \beta^*)$ on $(B^*)^c$ leading to a dominating negative term in the numerator. The stochastic term on the other hand can be controlled by assuming $y - \mathbb{E}y$ to be sub-Gaussian, in a very similar way the term $\ell_r(\beta) - \ell_r(\beta^*)$ is controlled in $B^*$.

The treatment of the denominator needs extra care to avoid any assumption on the growth of the true coefficients. Motivated by Atchadé (2017); Castillo & van der Vaart (2012); Castillo et al. (2015), we consider a Laplace-type prior ($\propto \exp\{-\kappa h\}$, where $h$ is Lipschitz outside a neighborhood around zero and $\kappa > 0$ is a constant) on the regression coefficients $\beta$. The right amount of tail thickness associated with such priors leads to an assumption free estimation of the regression coefficients $\beta$.

## 4 Discussion

Consider the general setup of Bayesian model selection (Bishop, 2006; Hoeting et al., 1999), where we are given a set of $K$ candidate models $\{\mathcal{M}_k\}_{k=1}^K$ with prior probabilities $\{p_k\}_{k=1}^K$. The $k$th model postulates a probability model $f_k(Y \mid \theta_k, \mathcal{M}_k)$ for the data with model parameters $\theta_k \in \Theta_k$, which is endowed with a prior $\pi_k(\cdot)$. Then, the posterior probability of the $k$th model given data $Y$ is given by

$$p_k(Y) := \frac{p_k m_k(Y)}{\sum_j p_j m_j(Y)}, \quad m_k(Y) = \int f_k(Y \mid \theta_k, \mathcal{M}_k)\, \pi_k(d\theta_k).$$

In particular, the maximum *a posteriori* model $\widehat{k} = \arg\max_k p_k(Y)$ reports the model with the highest posterior probability. Note that one can express $1/p_k(Y) = 1 + \sum_{j \neq k} (p_j/p_k)\, \mathrm{BF}_{jk}$, where $\mathrm{BF}_{jk}(Y) = \frac{m_j(Y)}{m_k(Y)}$ is the *Bayes factor* between models $j$ and $k$, which is simply the ratio of the marginal likelihoods. Asymptotic analysis of the posterior model probabilities necessarily require control over the behavior of $\log \mathrm{BF}_{jk}(Y)$ under the true data distribution, which may lie outside any of the candidate models considered. In such scenarios, our bounding technique can be generally applied.

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

## A  Appendix

## B  Proof of Theorem 1

Let $\mathcal{Y}_g$ denote the subset of the sample space $\mathcal{Y}$ where the events in Assumptions 1 and 2 both hold. We shall work inside the set $\mathcal{Y}_g$, with $\mathbb{P}(\mathcal{Y}_g) \geq 1 - \delta - \tilde{\delta}$ by Bonferroni's inequality.

We first prove the upper bound (4). By Assumption 1,

$$(1 - \eta) \leq \gamma(B^*) = \frac{\int_{B^*} e^{\ell(\theta, \theta^*)} \pi(\theta) d\theta}{\int_\Theta e^{\ell(\theta, \theta^*)} \pi(\theta) d\theta}.$$

Rearranging terms, this gives

$$\log \mathcal{Z}_\gamma \leq \ell(\theta^*) + \log\left(\frac{1}{1 - \eta}\right) + \log \int_{B^*} e^{\ell(\theta, \theta^*)} \pi(\theta) d\theta.$$

We now bound the integral in the right hand side of the above display. We have,

$$\int_{B^*} e^{\ell(\theta, \theta^*)} \pi(\theta) d\theta = \int_{B^*} e^{\ell_r(\theta) - \ell_r(\theta^*) + \mathbb{E}\ell(\theta, \theta^*)} \pi(\theta) d\theta \leq e^{Cd} \int_{B^*} e^{-(\theta - \theta^*)^{\mathrm{T}} H(\theta - \theta^*)/2} \pi(\theta) d\theta$$

$$\leq \{\sup_{\theta \in B^*} \pi(\theta)\} e^{Cd} (2\pi)^{d/2} |H|^{-1/2} \int_{B^*} \phi_d(\theta; \theta^*, H^{-1}) d\theta,$$

where $\phi_d(x; \mu, \Sigma)$ denotes a $d$-variate normal density with mean $\mu$ and covariance $\Sigma$ evaluated at $x \in \Re^d$. The bound (4) follows since $\int_{B^*} \phi_d(\theta; \theta^*, H^{-1}) d\theta = P(\|\xi\|^2 < Rd)$ for $\xi \sim \mathcal{N}_d(0, W^{1/2} H^{-1} W^{1/2})$.

For the lower bound, we use

$$\log \mathcal{Z}_f = \ell(\theta^*) + \int_\Theta e^{\ell(\theta, \theta^*)} \pi(\theta) d\theta \geq \ell(\theta^*) + \int_{B^*} e^{\ell(\theta, \theta^*)} \pi(\theta) d\theta.$$

We now bound the integral in the right hand side of the above display from below. We have,

$$\int_{B^*} e^{\ell(\theta, \theta^*)} \pi(\theta) d\theta = \int_{B^*} e^{\ell_r(\theta) - \ell_r(\theta^*) + \mathbb{E}\ell(\theta, \theta^*)} \pi(\theta) d\theta$$

$$\geq e^{-Cd} \int_{B^*} e^{-(\theta-\theta^*)^{\mathrm{T}} H(\theta-\theta^*)/(2c)} \pi(\theta) d\theta$$

$$\geq \left\{ \inf_{\theta \in B^*} \pi(\theta) \right\} e^{-Cd} (2\pi)^{d/2} c^{d/2} |H|^{-1/2} \int_{B^*} \phi_d(\theta; \theta^*, cH^{-1}) d\theta.$$

Finally, we have $\int_{B^*} \phi_d(\theta; \theta^*, cH^{-1}) d\theta = P(\|\xi\|^2 < Rc^{-1}d)$ where $\xi \sim \mathcal{N}_d(0, W^{1/2}H^{-1}W^{1/2})$.

## C    Posterior concentration in generalized linear models

Consider a generalized linear model with the canonical parameterization: $y_i \stackrel{\mathrm{ind.}}{\sim} P_{x_i^{\mathrm{T}}\beta}$ for $i = 1, \ldots, n$, and the log-likelihood as $L(\beta) := \log p_\beta(y) = \sum_{i=1}^n \left\{ y_i x_i^{\mathrm{T}} \beta - a(x_i^{\mathrm{T}} \beta) \right\}$, where $(y_i, x_i) \in \Re \times \Re^p$, $\beta \in \Re^p$ is the parameter of interest, and $a$ is a real valued convex function. We allow the true density $p_0(y)$ of $y_i$ to be misspecified and let $\mathbb{E}(y_i) = A_i$ and $\mathrm{Var}(Y) = \Sigma$. We note some important properties of the model.

### C.1    Properties of various aspects of the model

We define the pseudo-true parameter $\beta^*$ as $\beta^* = \arg \max_{\beta \in \mathbb{R}^p} EL(\beta)$. Under $P_{x_i^{\mathrm{T}}\beta^*}$, $\mathbb{E}(y_i) = a^{(1)}(x_i^{\mathrm{T}}\beta^*)$ and $\mathrm{Var}(y_i) = a^{(2)}(x_i^{\mathrm{T}}\beta^*)$. Also, $\nabla \mathbb{E}L(\beta^*) = 0$ which implies $\sum_{i=1}^n \{A_i - a^{(1)}(x_i^{\mathrm{T}}\beta^*)\} x_i = 0$. $V_0^2 := \mathrm{Var}\{\nabla L(\beta^*)\}$ and $D_0^2 := -\mathbb{E}\{\nabla^2 L(\beta^*)\} = X^{\mathrm{T}} W X$, where $W = \mathrm{diag}\{a^{(2)}(x_1^{\mathrm{T}}\beta^*), \ldots, a^{(2)}(x_n^{\mathrm{T}}\beta^*)\}$. Next, we look at some important divergences/distance measures defined as follows. A subscript 0 will indicate the divergence measure to be misspecified.

$$
\begin{aligned}
D_0(\beta^*, \beta) &:= \mathbb{E}\left\{ \log \frac{p_{\beta^*}(y)}{p_\beta(y)} \right\} = \sum_{i=1}^n \left\{ a(x_i^{\mathrm{T}}\beta) - a(x_i^{\mathrm{T}}\beta^*) - a^{(1)}(x_i^{\mathrm{T}}\beta^*) x_i^{\mathrm{T}}(\beta - \beta^*) \right\} \\
&= D(\beta^*, \beta), \\
V_0(\beta^*, \beta) &:= \mathbb{E}\left\{ \log \frac{p_{\beta^*}(y)}{p_\beta(y)} - D_0(\beta^*, \beta) \right\}^2, \\
D_{0,\alpha}(\beta^*, \beta) &:= \frac{1}{\alpha - 1} \log A_{0,\alpha}(\beta^*, \beta) := \frac{1}{\alpha - 1} \log \int \left\{ \frac{p_\beta(y)}{p_{\beta^*}(y)} \right\}^\alpha p_0(y) dy, \\
A_{0,\alpha}(\beta^*, \beta) &= \mathbb{E} \exp\left[ \alpha \langle y - \mathbb{E}y, X(\beta - \beta^*) \rangle \right] \exp\{-\alpha D(\beta^*, \beta)\}.
\end{aligned}
$$

Note that we define the misspecified divergences only for the pair $\beta^*, \beta$ which forces $D_0(\beta^*, \beta) \geq 0$. $D_{0,\alpha}(\beta^*, \beta)$ is not necessarily a divergence and we shall impose assumptions on the true distribution of $y_i$ which allows $D_{0,\alpha}(\beta^*, \beta) \geq 0$. For any $\beta_1, \beta_2$, $H^2(\beta_1, \beta_2) := 1 - A_{1/2}(\beta_1, \beta_2)$. Noting that

$$\log \frac{p_\beta^*(y)}{p_\beta(y)} = (\beta - \beta^*)^{\mathrm{T}} X^{\mathrm{T}} Y - \sum_{i=1}^n [a(x_i^{\mathrm{T}}\beta) - a(x_i^{\mathrm{T}}\beta^*)]$$

and $Var(Y) = \Sigma$, we have $V_0(\beta^*, \beta) \leq (\beta - \beta^*)^{\mathrm{T}} X^{\mathrm{T}} \Sigma X(\beta - \beta^*)$. Note that $D_0(\beta^*, \beta) \leq K(\beta, \beta^*) n \|\beta^* - \beta\|^2$, where $K(\beta, \beta^*) = \sup_{\tilde{\beta} \in L(\beta^*, \beta)} \lambda_p \{X^{\mathrm{T}} W(\tilde{\beta}) X/n\}$,
$W(\tilde{\beta}) = \mathrm{diag}\{a^{(2)}(x_1^{\mathrm{T}}\tilde{\beta}), \ldots, a^{(2)}(x_n^{\mathrm{T}}\tilde{\beta})\}$ and $L(\beta^*, \beta)$ is the line-segment connecting $\beta^*$ and $\beta$.

### C.2    Assumptions on the generalized linear model

We set our model assumptions to control the log-likelihood ratio

$$\frac{p_\beta(y)}{p_{\beta^*}(y)} = \exp\left[ \langle y - \mathbb{E}y, X(\beta - \beta^*) \rangle - D(\beta^*, \beta) \right]. \tag{7}$$

The first part in the right hand side of (7) is a stochastic term which can be controlled using appropriate sub-Gaussian assumption on $y - \mathbb{E}y$. The second term involves a deterministic quantity which can be bounded using an appropriate condition on the second derivative of the $a$ function. The following assumptions achieve this in a concrete fashion.

**Assumption 5.** *Let $\kappa_1 := \lambda_1(X^{\mathrm{T}} W X / n) > 0$.*

**Assumption 6.** *Assume $a$ satisfies $a(t + h) \geq a(t) + h\, a^{(1)}(t) + \mathrm{r}(|h|)\, a^{(2)}(t)/2$ for all $t, h$, where $\mathrm{r}(\cdot)$ is a rate function from $\mathbb{R}^+$ to $\mathbb{R}^+$ satisfying i) $\mathrm{r}(0) = 0$, ii) $\lim_{h \to 0} \mathrm{r}(h) = 0$ and iii) $r(h) \geq h^2/(\mathrm{r}_1 + \mathrm{r}_2 h)$ for $\mathrm{r}_1, \mathrm{r}_2 \geq 0$ not simultaneously 0.*

**Assumption 7.** *$K := \sup_{\beta : \|\beta - \beta^*\| \leq \epsilon_n} K(\beta, \beta^*) < \infty$ where $\epsilon_n$ is the rate of posterior convergence.*

**Remark 1.** *Assumption 6 can be used to provide a lower bound for $D(\beta^*, \beta)$ in the following manner. If $a$ satisfies Assumption 6,*

$$D(\beta^*, \beta) = \sum_{i=1}^{n} \left\{ a(x_i^{\mathrm{T}} \beta) - a(x_i^{\mathrm{T}} \beta^*) - a^{(1)}(x_i^{\mathrm{T}} \beta^*) x_i^{\mathrm{T}}(\beta - \beta^*) \right\}$$

$$\geq \sum_{i=1}^{n} \mathrm{r}(|x_i^{\mathrm{T}}(\beta - \beta^*)|) a^{(2)}(x_i^{\mathrm{T}} \beta^*).$$

*Defining $\mathrm{k}(h) = h^2/\mathrm{r}(h)$,*

$$D(\beta^*, \beta) \geq (\beta - \beta^*)^{\mathrm{T}} \left[ \sum_{i=1}^{n} \frac{a^{(2)}(x_i^{\mathrm{T}} \beta^*)}{\mathrm{k}(|x_i^{\mathrm{T}}(\beta - \beta^*)|)} x_i x_i^{\mathrm{T}} \right] (\beta - \beta^*)$$

$$\geq \frac{(\beta - \beta^*)^{\mathrm{T}} X^{\mathrm{T}} W X (\beta - \beta^*)}{\mathrm{r}_1 + \mathrm{r}_2 \|X\|_\infty \sqrt{d} \|\beta - \beta^*\|},$$

*where the last inequality follows from the fact that $\mathrm{k}(h) \leq \mathrm{r}_1 + \mathrm{r}_2 h$ and $|x_i^{\mathrm{T}}(\beta - \beta^*)| \leq \|X\|_\infty \sqrt{d} \|\beta - \beta^*\|$.*

### C.3 Assumption on the data generating distribution

We assume that $(y - \mathbb{E}y)$ is a centered sub-Gaussian random vector.

**Assumption 8.** *Assume that there exists a constant $\tau > 0$ such that for any $v \in \mathbb{R}^n$,*

$$\mathbb{E} \exp \langle y - \mathbb{E}y, v \rangle \leq e^{\tau^2 \|v\|^2 / 2}.$$

**Remark 2.** *For example, if $y$ has a joint Gaussian distribution with covariance matrix $\Sigma$, then we may take $\tau = \|\Sigma\|_2$. Under this assumption, we revisit the quantity $\langle y - \mathbb{E}y, X(\beta - \beta^*) \rangle$. We claim the following: with probability at least $1 - e^{-d}$,*

$$|\langle y - \mathbb{E}y, X(\beta - \beta^*) \rangle| \leq \tau \|X\|_2 \sqrt{d} \|\beta - \beta^*\|, \ \forall \ \beta \in \mathbb{R}^d.$$

*Note that the probability statement is uniform in $\beta$. The proof uses a majorizing measure theorem (see Theorem 4.1 of Liaw et al. (2017)). To prepare for the proof, note first that for any $\beta \in \mathbb{R}^d$,*

$$|\langle y - \mathbb{E}y, X(\beta - \beta^*) \rangle| \leq \left( \sup_{u \in T} |\langle y - \mathbb{E}y, Xu \rangle| \right) \|\beta - \beta^*\|,$$

*with $T = \mathcal{S}^{d-1} \cup \{0_d\}$. Define a stochastic process $W_u = \langle y - \mathbb{E}y, Xu \rangle$ for $u \in T$. Note that*

$$\sup_{u \in T} |\langle y - \mathbb{E}y, Xu \rangle| = \sup_{u \in T} |W_u - W_0| \leq \sup_{u, \tilde{u} \in T} |W_u - W_{\tilde{u}}|.$$

*We shall invoke Theorem 4.1 to obtain a high probability bound to the quantity in the right most side of the above display. The process $W$ has sub-Gaussian increments. We have, for any $u, \tilde{u} \in T$,*

$$\mathbb{E} e^{\lambda(W_u - W_{\tilde{u}})} \leq e^{\lambda^2 \tau^2 \|Xu - X\tilde{u}\|^2 / 2} \leq e^{\lambda^2 \tau^2 \|X\|_2^2 \|u - \tilde{u}\|^2 / 2}.$$

*Hence, for any $u, \tilde{u} \in T$,*

$$\|W_u - W_{\tilde{u}}\|_{\psi_2} \leq \tau \|X\|_2 \|u - \tilde{u}\|_2.$$

*This implies the constant $M$ in their theorem can be taken as $M = \tau \|X\|_2$. Finally, note that $diam(T) \leq 2$ and the Gaussian width of $T$, $\mathbb{E} \sup_{x \in T} \langle g, x \rangle$ for $g \sim N(0, I_d)$, is of the order $\sqrt{d}$. The proof is completed by taking $u = \sqrt{d}$.*

**Assumption 9.** *There exists $\phi \in (0, 1)$ such that*

$$n \geq \frac{\tau \mathrm{r}_2 d \|X\|_2 \|X\|_\infty}{\phi \kappa_1}, \quad \frac{(1 - \phi)\mathrm{r}_1}{\phi(\mathrm{r}_2 \|X\|_\infty \sqrt{d})} \geq \epsilon_n.$$

Since $\|X\|_2 \leq \sqrt{n} \|X\|_\infty$, Assumption 9 allows $d$ to increase with $n$ at a rate slightly slower than $\sqrt{n}$.

### C.4 Assumptions on the prior

We assume that $\pi$ is a product of $d$ densities of the form $e^{-\kappa h}$, for a function $h$ satisfying for some constant $c > 0$,

$$|h(x) - h(y)| \leq D + D|x - y|, \forall x, y \in \mathbb{R},$$

for some constant $D > 0$. This covers Laplace and Student densities, which corresponds to uniformly Lipschitz $h$. It also covers other smooth densities with polynomial tails, and densities of the form $c_\alpha \exp\{-\kappa |x|^\alpha\}$ for some $\alpha \in (0, 1]$ which corresponds to Lipschitz $h$ outside a neighborhood of the origin. On the other hand the standard normal density is ruled out.

### C.5 Main result on posterior contraction

In the following, we state our main theorem on posterior contraction using the assumptions on the data generating process in §C.3, the model in §C.2 with the prior in §C.4.

**Theorem 2.** *Assume Assumption 8 on the data generation mechanism and Assumptions 7, 5, 9 and 6 on model and the prior assumptions in §C.4. Then there exists positive constants $C_1, C_2$, such that for any $\eta \in (0, 1)$ there exists $\delta = e^{-C_2 d}/\eta$ such that $\mathbb{P}\{\gamma(B^*) \geq 1 - \eta\} \geq 1 - \delta$, where the set $B^* = \{\beta : \|\beta - \beta^*\| \leq C_1 \sqrt{d/n}\}$.*

In Theorem 2 we make no sparsity assumptions on $\beta$ and let the dimension $d$ to increase with $n$ at a rate slightly slower than $\sqrt{n}$. Notably, the convergence rate we obtained is sharp minimax ($\sqrt{d/n}$) without any logarithmic term. Our non-asymptotic version of the Laplace approximation as Theorem 1 in the main document is valid for $d$ growing at this rate. This is in stark contrast with Shun & McCullagh (1995) who showed that the remainder terms of the Laplace approximation do not vanish unless $d^3/n \to 0$. This is due to difference in assumptions in the likelihood and the prior. Also our Laplace approximation does not require maximizing the likelihood as in Shun & McCullagh (1995), instead we evaluate the likelihood at the pseudo-true parameter $\theta^*$.

Another salient feature of our result is the absence of any assumption on the norm of $\beta$ which is possible due to the use of a heavier tailed prior distribution on $\beta$. We conjecture that the use of a Gaussian prior will lead to a degradation of the convergence rate unless the norm of the true coefficients is appropriately bounded.

### C.6 Proof of Theorem 2

We divide up the proof into two separate parts.
**Treatment of the denominator:** In the following, we lower bound the normalizing constant of the posterior distribution. The technical details are fairly standard modification of

**Lemma 1.** *Under Assumption 7 and assuming $\pi$ satisfies the assumption in §C.4, we have for any sequence of numbers $\epsilon_n$ going to 0,*

$$\int \frac{p_\beta(y)}{p_{\beta^*}(y)} \pi(\beta) d\beta \geq c_\kappa^d e^{-\kappa \sum_{j=1}^d \{h(\beta_j^*) + D\}} \int_{\|z\| \leq \epsilon_n} e^{-Kn\|z\|^2/2 - \kappa D \sum_{j=1}^d |z_j|} dz,$$

*where $c_\kappa$ is the normalizer of $\pi$ for $d = 1$.*

**Treatment of the numerator:** In this section, we assume Assumptions 8, 7, 5, 9 and 6 and the prior assumptions in §C.4 Define $\Omega_n$ be the set

$$\sup_{u \in T} |\langle y - \mathbb{E}y, Xu \rangle| \leq \tau \|X\|_2 \sqrt{d}.$$

We first detail our main result for test construction. Define a mapping from $p_\beta$ to the space of finite measures as

$$p_\beta \mapsto q_\beta := \frac{p_0}{p_{\beta^*}} p_\beta \mathbb{1}_{\Omega_n}$$

For any $\epsilon > 0$, define $B(\beta_1; \epsilon) = \{p_\beta : \|\beta - \beta_1\| < \epsilon\}$. Denote by $\text{conv}\{B(\beta_1; \epsilon)\}$ the convex hull of $B(\beta_1; \epsilon)$. Pick any $\beta_1$ such that $\|\beta_1 - \beta^*\| = r$. Then the following holds.

**Lemma 2.** *Assume Assumption 8 on the data generation mechanism and Assumptions 7, 5, 9 and 6 on model. Then for $r \geq \frac{(1-\phi)r_1}{\phi(r_2\|X\|_\infty \sqrt{d})}$, there exists measurable functions $0 \leq \Phi_{n,\beta_1} \leq 1$ such that for every $n \geq 1$*

$$\sup_{p_\beta \in \text{conv}\{B(\beta_1; r/2)\}} \mathbb{E}_0 \Phi_{n,\beta_1} + \mathbb{E}_{q_\beta}(1 - \Phi_{n,\beta_1}) \leq \exp\left\{-\frac{n\kappa_1(1-\phi)\alpha\|\beta_1 - \beta^*\|}{2r_2\|X\|_\infty \sqrt{d}}\right\}.$$

Now consider the following decomposition for any sequence of measurable functions $\tilde{\Phi}_n$ (functions of $y^{(n)}$),

$$\mathbb{E}_0 \gamma\{(B^*)^c\} \leq \mathbb{E}_0 \tilde{\Phi}_n + \mathbb{E}_0\{\gamma\{(B^*)^c\}\mathbb{1}_{\Omega_n}(1 - \tilde{\Phi}_n)\} + \mathbb{P}_0(\Omega_n^c), \tag{8}$$

where $\mathbb{P}_0(\Omega_n^c) \leq e^{-d}$ from Remark 2. Let $D(\beta^*) = c_\kappa^d e^{-\kappa \sum_{j=1}^d \{h(\beta_j^*)+D\}} \int e^{-K\|z\|^2/2 - \kappa D \sum_{j=1}^d |z_j|} dz$. Writing for fixed $M > 0$,

$$U := \{\beta : \|\beta - \beta^*\| > M\epsilon_n\} = \bigcup_{j=1}^\infty U_{j,n} \tag{9}$$

where $U_{j,n} = \{\beta : jM\epsilon_n < \|\beta - \beta^*\| < (j+1)M\epsilon_n\}$, the second term in the rhs of (8) can be further decomposed as

$$\mathbb{E}_0\{\gamma\{(B^*)^c\}\mathbb{1}_{\Omega_n}(1 - \tilde{\Phi}_n)\} \leq D(\beta^*)^{-1} \sum_{j=1}^\infty \int_{U_{j,n}} \mathbb{E}_0\left[\mathbb{1}_{\Omega_n}(1 - \tilde{\Phi}_n)\frac{p_\beta(y)}{p_{\beta^*}(y)}\right]\pi(\beta)d\beta. \tag{10}$$

Let $N_{j,n} := N(jM\epsilon_n/2, U_{j,n}, \|\cdot\|)$ denote the $jM\epsilon_n/2$-covering number of $U_{j,n}$ with respect to $\|\cdot\|$. For each $j \geq 1$, let $S_j$ be a maximal $jM\epsilon_n/2$-separated points in $U_{j,n}$ and for each point $\tilde{\beta}_k \in S_j$ we can construct a test function $\Phi_{n,\tilde{\beta}_k}$ as in Lemma 2, with $r = jM\epsilon_n$. Then we set $\tilde{\Phi}_n$ to

$$\tilde{\Phi}_n = \sup_{j \geq 1} \max_{\tilde{\beta}_k \in S_j} \Phi_{n,\tilde{\beta}_k}.$$

From Lemma 2 since $\frac{(1-\phi)r_1}{\phi(r_2\|X\|_\infty \sqrt{d})} \geq \epsilon_n$,

$$\mathbb{E}_0\{\tilde{\Phi}_n\} \leq \sum_{j=1} N_{j,n} \exp\left\{-\frac{n\kappa_1(1-\phi)\alpha jM\epsilon_n}{2r_2\|X\|_\infty \sqrt{d}}\right\},$$

$$\mathbb{E}_0\{\gamma(B^*)\mathbb{1}_{\Omega_n}(1 - \tilde{\Phi}_n)\} \leq \sum_{j=1}^\infty \left\{\frac{\Pi(U_{j,n})}{D(\beta^*)}\right\} \exp\left\{-\frac{n\kappa_1(1-\phi)\alpha jM\epsilon_n}{2r_2\|X\|_\infty \sqrt{d}}\right\}.$$

Clearly $N_{j,n} \leq 9^d$ and

$$\frac{\Pi(U_{j,n})}{D(\beta^*)} \leq I^{-1} \times e^{\kappa dD} \int_{U_{j,n}} e^{\kappa \sum_{j=1}^d \{h(\beta_j^*) - h(\beta_j)\}} d\beta$$

where $I = \int_{\|z\| \leq \epsilon_n} e^{-Kn\|z\|^2/2 - \kappa D \sum_{j=1}^d |z_j|} dz$. Also the assumption in §C.4 entails

$$\sum_{j=1}^d \{h(\beta_j^*) - h(\beta_j)\} \leq dD + D\|\beta - \beta^*\|_1 \leq dD + 2D\sqrt{d}\|\beta - \beta^*\|_2 - D\|\beta - \beta^*\|_1.$$

Hence

$$\begin{aligned}
\frac{\Pi(U_{j,n})}{D(\beta^*)} &\leq I e^{2\kappa dD + 2D\sqrt{d}(j+1)M\epsilon_n} \int_{U_{j,n}} e^{-D\|\beta-\beta^*\|_1} d\beta \\
&\leq (I_2^d/I) \exp\{2\kappa dD + 2D\sqrt{d}(j+1)M\epsilon_n\}.
\end{aligned} \tag{11}$$

where $I_2 = \int_{\mathbb{R}} e^{-D|x|} dx$. Note that

$$\begin{aligned}
I &\geq e^{-\sqrt{d}\epsilon_n} \int_{\|z\| \leq \epsilon_n} e^{-Kn\|z\|^2} dz = e^{-\sqrt{d}\epsilon_n} \int_0^{\epsilon_n} e^{-Knr^2} r^{d-1} dr \\
&= \frac{1}{2} \epsilon_n^d (Kn\epsilon_n^2)^{-d/2} \big[\Gamma(d/2) - \Gamma(d/2, Kn\epsilon_n^2)\big],
\end{aligned} \tag{12}$$

where $\Gamma(a, x)$ is the incomplete Gamma function defined by $\int_x^\infty t^{a-1} e^{-t} dt$. From (8)-(11), and noting from (12) that $I \geq \exp\{-cd \log n\}$ for some constant $c > 0$, we have

$$\mathbb{E}_0\big\{\gamma\{(B^*)^c\}\mathbb{1}_{\Omega_n}(1 - \tilde{\Phi}_n)\big\} \leq \sum_{j=1}^\infty (I_2^d/I) \exp\Big\{2\kappa dD + \tag{13}$$

$$2D\sqrt{d}(j+1)M\epsilon_n - \frac{n\kappa_1 \alpha j M\epsilon_n}{2\mathrm{r}_2\|X\|_\infty \sqrt{d}}\Big\} \tag{14}$$

$$\leq \sum_{j=1}^\infty \exp\Big\{-C_1 \frac{n\kappa_1(1-\phi)\alpha j M\epsilon_n}{\mathrm{r}_2\|X\|_\infty \sqrt{d}}\Big\} \tag{15}$$

and

$$\mathbb{E}_0 \tilde{\Phi}_n \leq \sum_{j=1}^\infty 9^d \exp\Big\{-\frac{n\kappa_1 \alpha j M\epsilon_n}{\mathrm{r}_2\|X\|_\infty \sqrt{d}}\Big\} \tag{16}$$

$$\leq \sum_{j=1}^\infty \exp\Big\{-C_2 \frac{n\kappa_1 \alpha j M\epsilon_n}{\mathrm{r}_2\|X\|_\infty \sqrt{d}}\Big\} \tag{17}$$

for some constants $C_1, C_2 > 0$, by Assumption 9. Hence $\mathbb{E}_0 \gamma(B^*) > 1 - e^{-Cd}$ for some constant $C > 0$. An application of Markov's inequality concludes the proof of Theorem 2.

## D   Some auxiliary results

### D.1   Proof of Lemma 2

Set $\lambda_d := \tau\|X\|_2\sqrt{d}$ and $U_r + \lambda_d := n\kappa_1/\{\mathrm{r}_2\|X\|_\infty\sqrt{d}\}$. Then $U_r + \lambda_d \geq (1-\phi)\frac{n\kappa_1}{\mathrm{r}_2\|X\|_\infty\sqrt{d}}$, then

$$\frac{(U_r + \lambda_d)\mathrm{r}_1}{n\kappa_1 - (U_r + \lambda_d)\sqrt{d}\mathrm{r}_2\|X\|_\infty} \geq \frac{(1-\phi)\mathrm{r}_1}{\phi(\mathrm{r}_2\|X\|_\infty\sqrt{d})} = L_r$$

Due to Assumption 9, $U_r \geq \frac{(1-\phi)n\kappa_1}{\mathrm{r}_2\|X\|_\infty\sqrt{d}}$ and hence for $x \geq L_r$,

$$\lambda_d - \frac{n\kappa_1 x}{\mathrm{r}_1 + \mathrm{r}_2\|X\|_\infty\sqrt{d}x} < -(1-\phi)\frac{n\kappa_1 x}{\mathrm{r}_2\|X\|_\infty\sqrt{d}}.$$

Then from Remark 2, it follows if $\|\beta - \beta^*\| > L_r$,

$$\int_{\Omega_n} \left(\frac{p_\beta}{p_{\beta^*}}\right)^\alpha p_0 dy \leq \exp\left\{\alpha\lambda_d\|\beta-\beta^*\| - \frac{n\kappa_1\alpha\|\beta-\beta^*\|^2}{\mathrm{r}_1 + \mathrm{r}_2\|X\|_\infty\sqrt{d}\|\beta-\beta^*\|}\right\}$$

$$\leq \exp\left\{-\frac{n\kappa_1\alpha(1-\phi)\|\beta-\beta^*\|}{r_2\|X\|_\infty\sqrt{d}}\right\}.$$

By Theorem 6.1 of Kleijn & van der Vaart (2006), there exists test functions $\Phi_{n,\beta_1}$ such that for every $n \geq 1$

$$
\begin{aligned}
\sup_{p_\beta\in\text{conv}\{B(\beta_1;r/2)\}}\mathbb{E}_0\Phi_{n,\beta_1}+\mathbb{E}_{q_\beta}(1-\Phi_{n,\beta_1}) &\leq \sup_{p_\beta\in\text{conv}\{B(\beta_1;r/2)\}}\int_{\Omega_n}\left(\frac{p_\beta}{p_{\beta^*}}\right)^\alpha p_0 dy\\
&\leq \exp\left\{-\frac{n\kappa_1\alpha(1-\phi)\|\beta_1-\beta^*\|}{2r_2\|X\|_\infty\sqrt{d}}\right\}.
\end{aligned}
$$

$\square$

## D.2 Proof of Lemma 1

The proof follows along the lines of Lemma 11 of Atchadé (2017). We have,

$$
\begin{aligned}
\int\frac{p_\beta(y)}{p_{\beta^*}(y)}\pi(\beta)d\beta &= c_\kappa^d\int\frac{p_\beta(y)}{p_0(y)}e^{-\kappa\sum_{j=1}^d h(\beta_j)}d\beta\\
&\geq c_\kappa^d\int_{\beta:\|\beta-\beta^*\|\leq\epsilon_n}e^{\left\{\sum_{i=1}^n(y_i-\mathbb{E}y_i)x_i^{\mathrm{T}}(\beta-\beta^*)-Kn\|\beta-\beta^*\|^2/2-\kappa\sum_{j=1}^d h(\beta_j)\right\}}d\beta.
\end{aligned}
$$

Substituting $z = \beta - \beta^*$, we obtain

$$
\begin{aligned}
\int\frac{p_\beta(y)}{p_{\beta^*}(y)}\pi(\beta) &\geq c_\kappa^d\int_{\|z\|\leq\epsilon_n}e^{\left\{\sum_{i=1}^n(y_i-\mathbb{E}y_i)x_i^{\mathrm{T}}z-Kn\|z\|^2/2-\kappa\sum_{j=1}^d h(z_j+\beta_j^*)\right\}}dz\\
&\geq c_\kappa^d e^{-\kappa\sum_{j=1}^d\{h(\beta_j^*)+D\})}\int_{\|z\|\leq\epsilon_n}e^{\left\{\sum_{i=1}^n(y_i-\mathbb{E}y_i)x_i^{\mathrm{T}}z-Kn\|z\|^2/2-\kappa D\sum_{j=1}^d|z_j|\right\}}dz\\
&\geq c_\kappa^d e^{-\kappa\sum_{j=1}^d\{h(\beta_j^*)+D\}}\int_{\|z\|\leq\epsilon_n}e^{-Kn\|z\|^2/2-\kappa D\sum_{j=1}^d|z_j|}dz \qquad (18)\\
&\quad\times\exp\int_{\|z\|\leq\epsilon_n}\left\{\sum_{i=1}^n(y_i-\mathbb{E}y_i)x_i^{\mathrm{T}}z\right\}\tilde\pi(z)dz, \qquad (19)
\end{aligned}
$$

where the second last inequality follows from the fact that $h(z_j+\beta_j^*)\leq h(\beta_j^*)+D|z_j|+D$. The last inequality follows from an application of Jensen's where the expectation is taken with respect to $\tilde\pi$ given by

$$\tilde\pi(z)=\frac{e^{-Kn\|z\|^2/2-\kappa D\sum_{j=1}^d|z_j|}}{\int_{\|z\|\leq\epsilon_n}e^{-Kn\|z\|^2/2-\kappa D\sum_{j=1}^d|z_j|}dz}\mathbb{1}_{\|z\|\leq\epsilon_n}.$$

Noting that the integrand in (19) is an odd function, we have obtained the final result.

