# OpenReview forum: "Nonasymptotic Laplace approximation under model misspecification"
_TMLR — Rejected by TMLR_

### Review · Reviewer_h9oA · 2024-05-01

**Summary Of Contributions:**

This paper studies the Laplace approximation under misspecification in a non-asymptotic manner. The authors derive the Laplace approximation without Gaussianity of the posterior. In particular, they obtain a non-asymptotic two-sided bound on the likelihood.

**Audience:**

Yes

**Claims And Evidence:**

No

**Requested Changes:**

As mentioned above, the paper requires a heavy proof-read.

I think the authors could motivate a bit more why this problem is interesting, and maybe add a few more examples. Also, mention earlier that they verify their assumptions for a generalised linear model in section 3.

The presentation is clumsy and some letters are used repeatedly for different things. I will highlight some examples below:

The first 5 lines of the introduction are clumsy to read; the authors suppose $Y$ to be some data but don't define the sample space accordingly. The prob. dist. depends on $\theta$ but it is not mention that it lies in $\Theta$. Please use \mathbb{R} for real numbers. Say that $d\geq1$. Please use \mathrm{d}  instead of the plain $d$. What does proper prior means? Please specify this. Write explicitly the space the prior and posterior are coming from and to.

\mathbb{P} is used for both the prob. and for the true prob. distribution.

In eq. (2) use $KL$ for the Kullback-Leibler distance instead of D; moreover D is used too many times for different things in the appendix.

First line after (2), use \mathbb{E} and not plain E for expectation.

Maybe use $\theta^*$ for the true parameter and then use $\tilde{\theta}^{*}$ as the pseudo-true parameter.

Just below: "Throughout $C$, $C_{1}$, $C_{2}$,.. denote global postive constants." No please define them when you need them! (As in Ass. 2).

Ass. 3: proper define the prior.

Ass. 4. Why do the probability not depend on W and R?

On the bottom on p. 2: Is it obvious that $\delta\mathbb{E} l(\theta^{*})=0$?

P. 3: "Refer to Talagrand..." please rewrite.

P. 3: "We prove a general....in the Appendix ???". In addition to this then the appendix is wrongly formatted.

In Thm. 1, $\log P...$ does this mean prob.? Also, here define that $C_1$ and $C_2$ should be positive.

You mention that the bound in terms of $l(\hat{\theta})$ can be found in the appendix?

Start of section 3: You write covariate-response pairs but mention $y_{I}$ before $x_{I}$. Define $X$ with math. Use \mathbb{R} for real numbers.

Is it obvious that eq. (6) hold?

"display utilizes" - what does this mean?

You have have one transpose to much.

Please use another notation of $a^{(1)}$ and $a^{(2)}$

Please related and redefine \mathcal{T} and \mathcal{S}^{d-1}, such that it is clear \mathcal{S} is in \mathcal{T}. Moreover, you use plain T in the appendix.

$\lVert \cdot \rVert_2$ for operator norm? Use $\lVert \cdot \rVert_{\text{op}}$.

Please use [] after expectation and variance, and () after prob.

You write $\min_i g_i := \bar{g}^{-1}$ but you mean  $\min_i g_i =: \bar{g}^{-1}$ ?

Just before 3.2, you say "by setting $x=d$" but what about $\delta$?

In section 3.2: Theorem 1 in pg. 1493. ???

Define $v_{\max}=\max_i v_i$. Also, say that $I in [n]$.

The verification of Ass. 3 and 4 where the authors define the class of a functions and the rate function is messy to read. But they use \mathbb{R}.

Appendix is wrongly formatted. Starting with Appendix A (which is empty) and then Appendix B contains the proof of Thm 1. I won't go into detail of the appendices, but they are poorly written and the does not follow the main paper.

**Strengths And Weaknesses:**

Understanding the misspecified setting is fundamental. I think the authors could motivate this even more, e.g., make more examples and verifications of their assumptions. By this they could broaden their scope even more. The problem is interesting, but the paper is burdened by (very) many errors, which makes it difficult to understand and follow. In particular, the paper is badly written, the presentation is filled with errors, there is no consistency, and their notation is clumsy; I will give some examples under "Requested Changes".

---

> ### Author Response · Authors · 2024-05-19
> **Response to Reviewer 1 (h9oA)**
>
> Thanks for your comments on the article, and finding the work interesting. We respond to the central themes of your comments below:
>
> **Motivation:** The primary usage of our bound is to theoretically establish model selection consistency in situations where the true data generating model may lie outside the class of models entertained by the practitioner. Bayesian model selection weights each candidate model with a probability proportional to the prior probability assigned to the model times the marginal likelihood (or evidence) of said model. To establish frequentist properties of Bayesian model selection [e.g. to show that the posterior probability of one of the candidate models converges to one (with high probability under the true model)], one thus needs to analyze the behavior of the marginal likelihood under a given model under the true data generating distribution. This is the context where our bound becomes helpful, because it establishes two-sided bounds on the log-marginal likelihood with high probability under the true data generating model. We achieve this without the need to prove an asymptotic normality result for the posterior distribution (Bernstein–von Mises theorem) under each candidate model. Thus the main contribution is the ease of applicability across a wide-range of model selection problems. We mention this briefly in the discussion section, and given a chance to revise, we would elaborate that discussion and add it to the introduction.
>
> **Examples:** We will add a median (or quantile) regression example to illustrate these points, and simultaneously the need for separating the deterministic and stochastic components of the log-likelihood ratio (to respond to a question raised by R3). For quantile regression (using an asymmetric Laplace working model; symmetric Laplace for median regression), posterior consistency has been established in the Sriram et al. paper cited in our manuscript, allowing verification of Assumption 4. For median regression, verification of Assumptions 1 and 2 can leverage results from Section 5.3 of Spokoiny et al. (Spokoiny, Vladimir. "Parametric estimation. Finite sample theory." (2012): 2877-2909). In particular, their Lemma 5.9 offers a quadratic expansion of the expected log-likelihood ratio, which can be readily used to verify Assumption 2. Extending this to the quantile regression framework should be possible.
>
> **Typographical issues:** We will address the typographical issues provided the opportunity to revise.

---

### Review · Reviewer_dR8n · 2024-05-10

**Summary Of Contributions:**

The authors propose a non-asymptotic bound for the Laplace approximation applied to estimate the data marginal log likelihood. This bound is valid in a high-dimensional case where the dimension is growing with the sample size $n$ (at a rate slower than $\sqrt{n}$) and in the mis-specified setting where the data likelihood does not match the data generating process. The main and only result of this paper is Theorem 1, giving the upper and lower bounds on the approximation, in terms of the prior and the CDF of a normal variable.

**Audience:**

Yes

**Broader Impact Concerns:**

I have no concern on broader impact.

**Claims And Evidence:**

Yes

**Requested Changes:**

- Firstly, I suggest adding specific examples to illustrate the bound of Theorem 1.
- Secondly, I also suggest to refine the literature review, in particular with respect to the assumptions for the validity of the Laplace approximation (Rossell & Rubio 2018), and maybe adding the recent references on non-asymptotic bounds (Spokoiny 2023, Helin et al. 2022)
- Finally, the difficulty of the extension of the posterior concentration result should be discussed, and the discussion/conclusion section should be improved.

**Strengths And Weaknesses:**

Strengths:
- The literature on the non-asymptotic properties of the Laplace approximation for Bayesian inference is not very large and to my knowledge, mainly contains bounds on the total variation distance between the posterior distribution and the approximated Gaussian distribution (Helin et al., 2022, Spokoiny 2023). In contrast, this paper provides bounds on the marginal data log likelihood that are valid in quite general settings.
- The paper is generally very well written, the concepts and assumptions are well explained and its arguments are correct.

Weaknesses:
- The result, although elegant and valid in a wide range of settings, does not seem usable as it is, since the bound depends on the neighborhood $B^\star$  around the optimal parameter, and the choices of matrix $W$ and constant $R$ seem to depend intricately on how one can verify the assumptions. It would be insightful if the authors could provide examples on how to use the bound in specific models, and in particular, on the choice for $W$ and $R$.
- More generally, in my opinion, the paper does not provide enough illustration of the result with specific statistical models and priors, nor enough comparison of the assumptions with the existing literature. For instance, how much more general are the assumptions here compared to the ones of Section S10 in Rossell & Rubio (2018), on the validity of the Laplace approximation for mis-specified models?
- The discussion section (Section 4) seems a bit out-of-place as it is almost an introductory paragraph on the utility of the Laplace approximation in Bayesian model selection (for computing Bayes factors). For me, this paragraph would better fit in the introduction.
- In the proof of posterior concentration with generalised linear models in misspecified settings (Appendix C), it is not clear if it is a direct extension of Atchade (2017) or if there is any technical difficulty that the paper addresses. It seems that the technique is quite standard but I may be missing something.


Spokoiny, V. (2023). Dimension Free Nonasymptotic Bounds on the Accuracy of High-Dimensional Laplace Approximation. SIAM/ASA Journal on Uncertainty Quantification, 11(3), 1044-1068

Helin, T., & Kretschmann, R. (2022). Non-asymptotic error estimates for the Laplace approximation in Bayesian inverse problems. Numerische Mathematik, 150(2), 521-549.

---

> ### Author Response · Authors · 2024-05-19
> **Response to Reviewer 2 (dR8n)**
>
> We thank you for the careful reading of the article, your thoughtful and encouraging comments, and suggesting relevant references. We respond to the central themes of your comments below:
>
> **Choices of W and R and illustration in examples:** Thanks for this comment.  R is intrinsically tied to the radius of the neighborhood associated with the posterior contraction.  For parametric problems, if $W$ is taken to be the identity matrix, $R$ is of the order $1/n$.  This is clarified in Theorem 2 for the generalized linear models. We agree that additional concrete examples will help fix these ideas. *We will add a median (or quantile) regression example to illustrate these points, and simultaneously the need for separating the deterministic and stochastic components of the log-likelihood ratio (to respond to a question raised by R3).* For quantile regression (using an asymmetric Laplace working model; symmetric Laplace for median regression), posterior consistency has been established in the Sriram et al. paper cited in our manuscript, allowing verification of Assumption 4. For median regression, verification of Assumptions 1 and 2 can leverage results from Section 5.3 of Spokoiny et al. (Spokoiny, Vladimir. "Parametric estimation. Finite sample theory." (2012): 2877-2909). In particular, their Lemma 5.9 offers a quadratic expansion of the expected log-likelihood ratio, which can be readily used to verify Assumption 2. Extending this to the quantile regression framework should be possible.
>
> **Refining literature review:** Thanks for the suggested literature. *We will cite the two papers and discuss them. We will also compare our assumptions to Rossell & Rubio, who we already cite.* They work with a specific model (asymmetric Laplace), and exploit log-concavity and asymptotic normality. Our approach instead relies on controlling local fluctuations of the stochastic part of the log-likelihood ratio via concentration inequalities.
>
> **Regarding bringing the discussion to the forefront:** This is an excellent suggestion. *Given an opportunity to revise, we would elaborate the discussion section and add it to the introduction.* This will help provide necessary context to the usage of our bound, i.e., to establish model-selection consistency results in scenarios where the true model may be outside the class of models entertained by the practitioner. We also discuss this in detail in our first bulleted response to R3.
>
> **Comment regarding posterior contraction for generalized linear models:** Thanks for this comment. Our proof follows the general strategy (based on constructing test functions) for proving posterior concentration, borrowing additional ingredients from Atchade’s work. Section 3 of Atchade (2017) deals with posterior contraction in high dimensional sparse Bayesian logistic regression.  As our focus is on generalized linear models without imposing sparsity restrictions (thereby allowing dense models), we cannot directly use their result, although there are some commonalities in the proof technique. A key difference  is to exploit our Assumption 6 to control log-partition functions for general exponential family models, while Atchade (2017) considers the log-partition function specifically for a  logistic likelihood. *We will add this discussion.*

---

### Review · Reviewer_a5TH · 2024-05-12

**Summary Of Contributions:**

In this paper, the authors derive theoretical upper and lower bounds for approximating the model evidence. Specifically, the bounds can be viewed as a modification of the Laplace approximation. Moreover, the proposed bounds also hold for model mismatch, which is more realistic.

**Audience:**

Yes

**Claims And Evidence:**

Yes

**Requested Changes:**

- A section on how the bounds can be used in practice *Critical*
- Empirical evidence on the tightness of the bounds *Critical*
- A related works section *Critical*

**Strengths And Weaknesses:**

I want to preface this section by stating that I am not expert in this area so I apologize for any inaccuracies.

# Strengths

The biggest strength of the paper is its generality which is rather impressive. No assumptions are made on the form of the posterior distribution nor do the authors use large sample limits. Moreover, the fact that the bounds hold in the model mismatch case is very impressive!

# Weaknesses

There are couple of weaknesses that hold the paper back in my opinion. First, I'm worried about the applicability of the bounds. To start, while it is impressive that the derived bounds are a slight modification of the commonly used Laplace approximation, it is not clear how the bounds can benefit the practitioner. Specifically, the bounds are a function of $\log \vert H \vert$ while the the Laplace approximation uses $-d \log n / 2$, While the authors mention that the determinant of $H$ is very close to $n^{-d/2}$, how does using that approximation affect the bounds in practice? Next, it would be great if there was some discussion of the tightness of these bounds. Looking at equations 4 and 5, the tightness of the bounds seems to be dominated by $C_1$ and $C_2$ which can be arbitrarily large. It would have been great if there were some empirical experiments demonstrating the tightness of these bounds. Lastly, I think a brief section on how the bounds can be made practical would be great. For instance, how could a practitioner obtain a rough estimate of $C_1$ and $C_2$.

Second, I think that the text can be made more readable. For instance, before Assumptions 1-4 were stated it would have been great if that authors motivated the choice of rewriting the log likelihood ration as a sum of a stochastic and deterministic components. Another example, is that on page $\hat{\theta}$ is used for the first time without any introduction (moreover, it is not clear what it means).

Lastly, I think it would be great to discuss other methods for sandwiching the log marginal likelihood. A work that comes to mind is: https://arxiv.org/pdf/1511.02543.

---

> ### Author Response · Authors · 2024-05-19
> **Response to Reviewer 3 (a5TH)**
>
> Thank you for going over the article, and your thoughtful comments and feedback. We respond to the central themes of your comments below:
>
> **Regarding applicability of the bounds:** We would like to clarify here that the primary usage of our bound is to theoretically establish model selection consistency in situations where the true data generating model may lie outside the class of models entertained by the practitioner. Bayesian model selection weights each candidate model with a probability proportional to the prior probability assigned to the model times the marginal likelihood (or evidence) of said model. To establish frequentist properties of Bayesian model selection [e.g. to show that the posterior probability of one of the candidate models converges to one (with high probability under the true model)], one thus needs to analyze the behavior of the marginal likelihood under a given model under the true data generating distribution. This is the context where our bound becomes helpful, because it establishes two-sided bounds on the log-marginal likelihood with high probability under the true data generating model. We achieve this without the need to prove an asymptotic normality result for the posterior distribution (Bernstein–von Mises theorem) under each candidate model. Thus the main contribution is the ease of applicability across a wide-range of model selection problems. *We mention this briefly in the discussion section, and given a chance to revise, we would elaborate that discussion and add it to the introduction.*
>
> **Regarding estimates of C_1 and C_2:** Although a full characterization of the constants $C_1$ and $C_2$ is not needed for most practical applications (such as model selection consistency), it is definitely possible to give very accurate expressions for the constants $C_1$ and $C_2$ which are expressed in terms of a set of constants $c$ and $C$ appearing in Assumptions 1 and 2 respectively. We reiterate that even in the misspecified setting, the constants $c$ and $C$ can be obtained from the properties of the likelihood itself and do not depend on the data generation mechanism. More specifically, for regression problems, $c$ can be expressed as a constant multiple of the gram matrix associated with the design.  In addition,  $C$ can be expressed in terms of the fluctuation of $l_r(\theta) - l_r(\theta^*)$ as a multiple of $d$. A sharp estimate of $C$ (independent of $\theta^*$) can also be obtained by applying an appropriate concentration inequality. *We shall add this discussion about the choice of the constants.*
>
> **About improving readability:** Thanks for this point. In particular, the comment about decomposing into deterministic and stochastic parts is highly relevant, and we would like to clarify this further. It is important to separate into deterministic and stochastic components because in many interesting examples, the log-likelihood itself may not admit a quadratic expansion, but the expected log-likelihood does. For example, in median or quantile regression, the log-likelihood is not smooth, however, the expected log-likelihood is smooth, and admits a quadratic expansion. Our approach is well-suited to exploit this. *We will add this discussion, along with a quantile regression example, to the document.* Please also see the first bullet point in response to R2 for additional details regarding this model.
>
> **About related work:** Thanks for the reference. Since our primary goal is theoretical, we did not discuss the extensive literature on numerical approximations to the marginal likelihood in the previous version. *We will be happy to add a related work section, discussing the cited paper and related literature on numerically sandwiching the marginal likelihood.*

---

### Decision · Action_Editor_cTce · 2024-06-18

**Recommendation:** Reject

**Comment:**

The reviewers were clearly impressed by the theoretical result (Theorem 1), but they also raised many problems. While they agree that this result should ultimately be published, they were all reluctant to accept the paper without having a chance to review the revised version (as would be the case if I would accept the paper with minor revision). Thus, after a discussion, they all recommended to reject the paper in its current form, and to strongly encourage the authors to revise the paper and resubmit it. Note that there is no "major revision" option in TMLR, but in this case, the "reject with encouragement to resubmit" should be interpreted as a "major revision" decision.

The main problem raised by the reviewers is that the results might be difficult to apply in practice [dR8n]. Thus they requested example of application to standard statistical models [a5TH,dR8n,h9oA], together with a comparison to existing results in this model [dR8n]. Note that [h9oA] proposes a generalized linear model, which would be nice, but of course other choices are possible.

While reviewer [dR8n] finds the paper well written at a high level, the huge number of typos and minor mistakes is another important problem to fix [h9oA].

Other points were raised by the revewiers, that could also be discussed in the paper:
- possibility of extension to concentration of Laplace approximation [dR8n],
- empirical evidence of the tightness of the bound [a5TH],
- missing related works, see the paper mentioned by [a5TH] and [dR8n]. In the line of the works mentioned by [dR8n] I would also ask the authors to check
1) Kasprzak, Mikołaj J., Ryan Giordano, and Tamara Broderick. "How good is your Laplace approximation of the Bayesian posterior? Finite-sample computable error bounds for a variety of useful divergences." arXiv preprint arXiv:2209.14992 (2022).
2) Katsevich, Anya. "Tight dimension dependence of the Laplace approximation." arXiv preprint arXiv:2305.17604 (2023).

I wish the authors good luck with the revision of the paper.

**Audience:**

While the main target are the theoreticians of Bayesian inference, the result is of interest to all researchers using Bayesian machine learning in practice.

**Claims And Evidence:**

The authors prove non-asymptotic upper and lower bound on the log-marginal likelihood, under assumptions that are far more general than the ones required in standard analysis of the Laplace approximation. As a consequence, we have non-asymptotic results on the Laplace approximation, that hold under model misspecification.

**Resubmission Of Major Revision:**

The authors may consider submitting a major revision at a later time.